# Synthesize high-dimensional longitudinal electronic health records via hierarchical autoregressive language model

Brandon Theodorou[1,2], Cao Xiao[2] & Jimeng Sun [1,2] ✉

Synthetic electronic health records (EHRs) that are both realistic and privacy-preserving offer alternatives to real EHRs for machine learning (ML) and statistical analysis. However, generating high-fidelity EHR data in its original, high-dimensional form poses challenges for existing methods. We propose Hierarchical Autoregressive Language mOdel (HALO) for generating longitudinal, high-dimensional EHR, which preserve the statistical properties of real EHRs and can train accurate ML models without privacy concerns. HALO generates a probability density function over medical codes, clinical visits, and patient records, allowing for generating realistic EHR data without requiring variable selection or aggregation. Extensive experiments demonstrated that HALO can generate high-fidelity data with high-dimensional disease code probabilities closely mirroring (above 0.9 $R^2$ correlation) real EHR data. HALO also enhances the accuracy of predictive modeling and enables downstream ML models to attain similar accuracy as models trained on genuine data.

The widespread adoption of electronic health record (EHR) systems has established the foundation for machine learning (ML) and artificial intelligence (AI) applications in healthcare. The EHR data is highly complex, comprising over 10,000 unique medical codes for diagnoses, procedures, and medications, as well as thousands of lab measurements. Each patient record can include multiple visits with combinations of diagnoses, procedures, medications, and labs. These combinations create intricate relationships and complex patterns across tens of thousands of medical codes. AI and ML techniques are used to learn and model complex patterns in EHR data, enabling applications such as clinical predictive modeling[1,2], health monitoring[3,4], computational phenotyping[5,6], treatment recommendations[7–9], and more. However, the progress of AI and ML in healthcare is often impeded by the difficulty of accessing and sharing large real EHR datasets. Sharing EHR data is challenging due to privacy, security, and legal constraints. While patient de-identification can alleviate some of these concerns by removing obvious patient identifiers such as name, address, and birth date[10,11], studies have shown that the risk of re-identification remains high even after thorough de-identification[12–14].

Using synthetic patient data can offer a safer alternative to sharing real EHR data. Generative models can produce synthetic datasets as substitutes for real patient data[15–21]. Various methods have been proposed in the literature, including structured patient record generation[19,20,22–24] and longitudinal record generation[15,16,21].

To date, existing methods cannot generate realistic EHR data in its original, high-dimensional form. The high dimensionality of EHR data, along with rare and sparse variables and complex relationships among variables, makes the generation task a difficult one. Consequently, existing approaches all concede to creating lower-dimensional data by either aggregating variables or using a subset of more common variables of a manageable size. For example, the MedGAN method[19] modeled 615 disease categories without longitudinal information; the SynTEG model[15] aggregates codes to higher level phenotypes and then removes rare phenotypes, resulting in only 1276 variables; the ehrM-GAN approach[21] reduced the variable dimension to be <100, and EVA[16] models frequent co-occurrence patterns in the original EHR data as one-hot vectors, limiting its ability to generate diverse and novel co-occurrence patterns. Our supplementary information provides a table of these dimensionalities of existing methods. While these

[1]University of Illinois at Urbana-Champaign, 201 North Goodwin Avenue, Urbana, IL, USA. [2]Medisyn Inc., Las Vegas, NV, USA. ✉e-mail: jimeng@illinois.edu

low-dimensional approaches may capture the proper statistics on a small number of variables and support narrow ML use cases relying solely on those variables, the resulting synthetic data is inadequate for broader applications that require high-dimensional data including comprehensive statistical analysis, patient phenotyping, billing prediction and analysis, disease staging, and comprehensive data sharing.

We propose an approach for generating high-dimensional EHR data in its native form: the Hierarchical Autoregressive Language Model (HALO). This model, shown in Fig. 1, takes an autoregressive and probabilistic approach and can capture the hierarchical distribution of EHR records and their temporal relationships. Using a hierarchical approach to model binary sequences of over a million variables, HALO can efficiently learn and represent complex patterns in EHR data.

HALO works by utilizing a pair of modules to represent both the visit- and code-level structures of a patient record. First, it uses a coarse, visit-level module to factorize the probability along each of a patient's visits and to efficiently process and represent a patient's past medical history. It then adds fine, code-level modeling to generate each variable in a given visit based on both that past history and also the previous variables in the same visits for maximum intra-visit cohesion.

We evaluate the performance of HALO by training it on a comprehensive outpatient claims dataset, as well as the MIMIC-III inpatient EHR data[25], and compare the results with a diverse set of existing synthetic EHR data generation techniques such as refs. 15,16,26.

We evaluate the data quality based on its utility in modeling the statistical data distribution and for supporting ML models. HALO can accurately synthesize high-dimensional EHR data via modeling disease code probabilities ($d \approx 10,000$), disease code co-occurrence probabilities within a visit ($d \approx 1,000,000$), and conditional probabilities across consecutive visits ($d \approx 5,000,000$). In our experiments, we found that HALO achieves a correlation coefficient of above $0.9 R^2$ when compared to real EHR data, demonstrating its ability to generate realistic data.

In addition to generating high-fidelity and granular EHR data, we show that HALO improves predictive modeling on our EHR dataset by more than 17% compared to the leading baseline. We evaluate the predictive accuracy and perplexity of HALO on a hold-off test set, demonstrating its superiority. Furthermore, the synthetic data generated by HALO enable downstream phenotyping ML models to achieve comparable accuracy to models trained on real data, with an AUC of 0.938 for HALO data versus 0.943 for real data. We then demonstrate that combining real and synthetic data generated by HALO can improve the accuracy of ML models even more compared to using just real EHR data. Furthermore, we show that HALO generates realistic data while simultaneously protecting patients' privacy in the training data, as evaluated by a series of privacy metrics.

## Results
### Problem formulation
Structured EHRs are multi-level longitudinal records, where each patient is represented by a sequence of visits. Each visit is characterized by a set of medical codes, reflecting the diagnoses, procedures, and medications administered during that visit. Additional patient information, such as demographics, disease phenotype labels, lab test results, and inter-visit time, can also be included. We begin by formalizing the problem and introducing key notations that will be used throughout.

**EHR data.** We represent a patient record $\mathcal{R}$ as a sequence of visits over time such that

$$\mathcal{R} = \mathcal{V}^{(1)}, \mathcal{V}^{(2)}, \cdots \mathcal{V}^{(T)} \tag{1}$$

where each visit $\mathcal{V}^{(t)}$ contains a varying number of medical codes $m_1^{(t)}, m_2^{(t)}, \cdots, m_{|\mathcal{V}_\mathcal{C}^{(t)}|}^{(t)} \in \mathcal{C}$, lab values $l_1^{(t)}, \cdots, l_{|\mathcal{V}_\mathcal{L}^{(t)}|}^{(t)} \in \mathcal{L}$, and the inter-visit time gap $g^{(t)}$. $\mathcal{C}$ is then the set of all medical codes in our vocabulary,

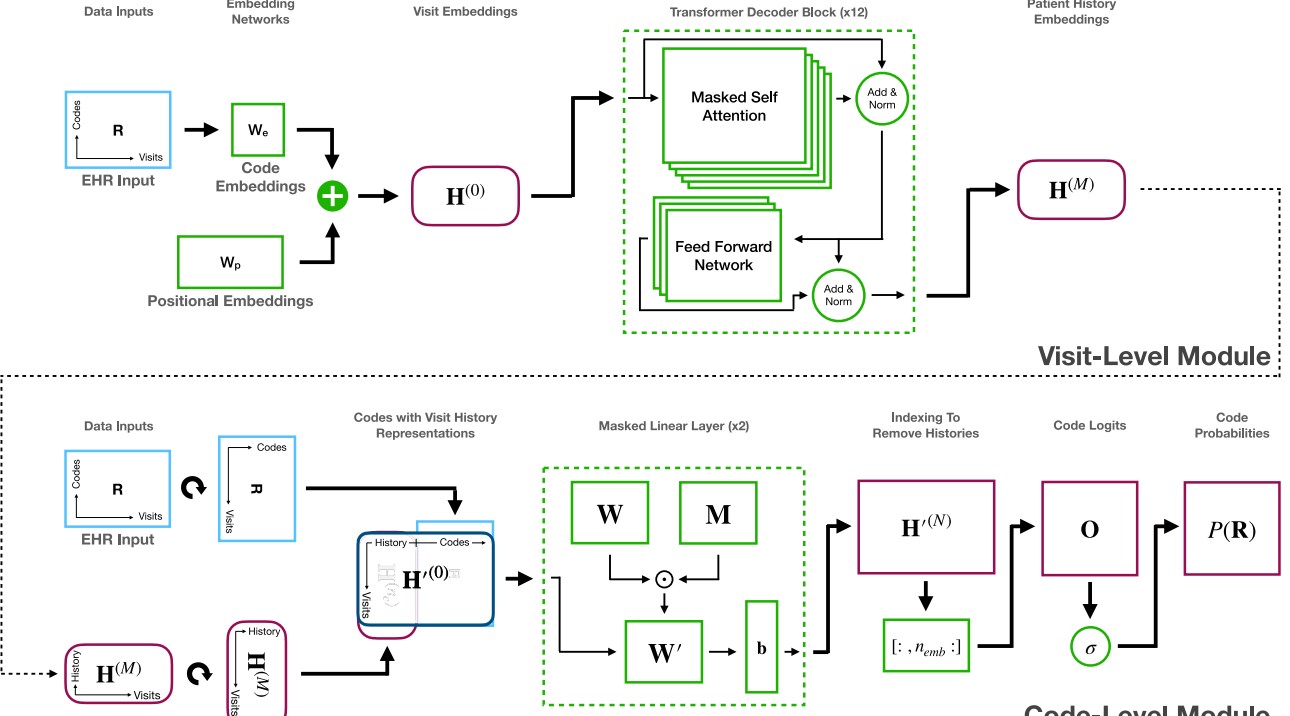

**Fig. 1 | The proposed HALO model.** The architecture of HALO utilizing an autoregressive multi-granularity approach which analyzes at both the visit and code level to generate next code probabilities based on the history of all previous visits as generated through a stack of transformer decoder layers and the previous codes in the current visit through a series of masked linear layers.

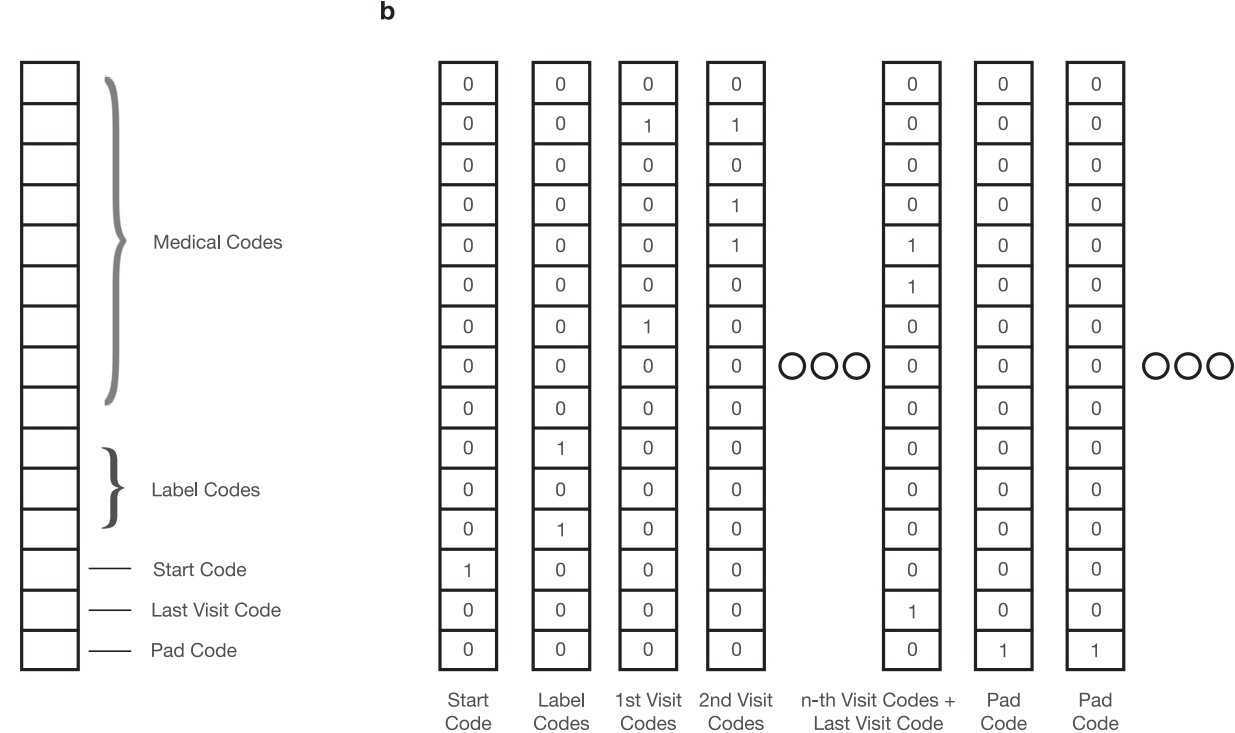

**Fig. 2 | The data formatting. a** The visit representation. Each visit is represented as a multi-hot vector containing indices for medical codes, static label codes to cover demographics and disease phenotypes, and special codes describing the shape and temporal ordering of the patient's visit. **b** The EHR representation. An EHR is then represented as a matrix constructed as a series of temporally ordered visit vectors.

including diagnoses, procedures, and medications and $\mathcal{L}$ is the set of all labs. Beyond the longitudinal records, a patient record also possesses some static information $\mathcal{S}$ containing demographics such as gender, race, and birth year and disease phenotype label $\mathcal{D}$ indicating major and persistent disease conditions.

**Matrix representation.** To allow input to `HALO` and other machine learning models, we then convert $\mathcal{R}$,$\mathcal{S}$, and $\mathcal{D}$ into a matrix representation $\mathbf{R}$. Specifically, we build $\mathbf{R} = [\mathbf{v}_s, \mathbf{v}_l, \mathbf{v}_1, \cdots, \mathbf{v}_T, \mathbf{v}_e]$, a matrix containing a sequence of the vector representations for each of the patient's $T$ visits, a preceding start visit, label visit, and a succeeding end visit.

The start visit $\mathbf{v}_s$ is a one-hot vector containing a special start code added to $\mathcal{C}$ to signify the start of the record often required for certain model architectures.

The label visit $\mathbf{v}_l$ similarly contains special codes added to $\mathcal{C}$ representing demographic and chronic disease phenotypes from $\mathcal{S}$ and $\mathcal{D}$, respectively. For example, this label visit will have codes representing the patient's gender, racial and ethnic groups, birth year, and any chronic labels.

Each subsequent visit $\mathbf{v}_t \in \mathbb{R}^{|\mathcal{C}|}$ is then represented as a multi-hot binary vector representing medical codes, lab values, and inter-visit gaps present in that visit. To represent continuous lab values and visit gaps in a discrete form, we employ a granular discretization. This is achieved by adding multiple range codes to $\mathcal{C}$ for each lab test and for the intervals between visits. By converting all medical information into binary variables, $c_t^i$ represents the presence of the $i$th code in $\mathcal{C}$ in the $t$th visit of the patient record $\mathcal{R}$.

Finally, to signal the end of the patient record in $\mathbf{v}_e$, a special last visit code is added to $\mathcal{C}$, serving a similar purpose to a stop token in natural language generation. This not only enables generative models to learn when to terminate records but also allows for $\mathbf{R}$ to be padded through additional columns into a constant length for batch input without altering its content.

Figure 2 depicts the format of the visit vector and the EHR representation, and we provide a table of notations for reference in our supplementary information.

**Generation task.** is to create $\mathbf{R}'$, a synthetic patient record that is statistically similar to and offers the utility of $\mathbf{R}$ without any one-to-one mapping to a real patient. Our `HALO` method does this by learning the distribution $P(\mathbf{R})$.

### Experimental design
We evaluate our method and compare it to several baselines comprising both recently proposed models and other logical autoregressive model architectures on a series of experiments on both outpatient and inpatient EHR datasets. To maintain the fidelity of the original EHR data, our experiments focus on synthesizing original granular medical codes without aggregating or combining codes. Specifically, we seek to answer the following questions.

- Is `HALO` effective at modeling the underlying data distribution of electronic health records?
- Can `HALO` produce a synthetic dataset that is statistically similar to real EHR data?
- Can `HALO` augment real data for more accurate disease phenotyping prediction?
- Can `HALO` generate realistic continuous variables such as lab results and visit time gap?
- Can `HALO` preserve patient privacy in the training?

### Datasets and experimental setup
**Datasets.** We use two datasets for our experiments:

(1) The outpatient EHR is from a large real-world US claims data. It contains 929,268 patients and binary labels for 11 chronic diseases (specific diseases and patient counts are included in the supplementary information). This yields a final real-world

outpatient EHR dataset with an average of 34.16 visits per record and 3.52 codes per visit with 9882 unique ICD-10 codes.

(2) The inpatient EHR is from the MIMIC-III ICU stay dataset[25]. It contains 46,520 patients with 25 disease phenotype labels as defined by the MIMIC benchmark[27]. This dataset has an average of 1.26 visits per record and 15.11 codes per visit with 6841 unique ICD-9 codes. Note that this includes patients with just a single visit (and as we will show, HALO's Code-Level Module allows it to be very effective on those patients).Both datasets share the same patient representation as a series of visits along with chronic disease phenotype labels. We keep the ICD codes in the data without code aggregation or removing any infrequent codes.

**Experiment setup.** We use a 0.8–0.2 training-test split with an additional 0.9–0.1 training-validation split during training for both outpatient and inpatient datasets. We use the Adam optimizer with a learning rate 1e−4 (which was arrived upon through experimentation). We use a batch size of 48 and train for 50 epochs, saving the model with the lowest loss on the validation set. We implement the model and train in the Python 3.6.9 coding language using the PyTorch 1.9.0+cu111 framework[28] along with the scikit-learn 0.24.2 and NumPy 1.17.2 packages. Finally, all experiments are done via one NVIDIA TESLA V100 GPU with 32 GB RAM. The HALO source code is publicly available on GitHub at https://github.com/btheodorou99/HALO_Inpatient.

## Baseline methods

Below we outline the baseline methods and the necessary alterations to those baselines to adapt to our problem setting.

- HALO-Coarse: This baseline is an ablation baseline consisting of just the coarse, visit-level granularity module of the full HALO architecture. It generates each code probability based on all previous visits (grouped into a multi-hot representation) but without the fine, inter-visit modeling such that $P(c_i^i)$ is modeled by $P(c_i^i|\mathbf{v}_1, \cdots, \mathbf{v}_{t-1})$ instead of $P(c_i^i|\mathbf{v}_1, \cdots, \mathbf{v}_{t-1}, c_i^1, \cdots, c_i^{i-1})$. It consists predominantly of 12 transformer decoder blocks in the model of Radford et al.[29] augmented to support multi-hot as opposed to one-hot inputs and outputs within the embedding layer and final activation layer.

- GPT model[29]: We applied the GPT model without any augmentation to support multi-hot inputs and outputs but instead with the conversion of EHRs to a fully one-hot sequential representation. However, this model had to be shrunk down to 3 blocks from 12 to fit into memory because this greatly expanded the length of the sequences.

- LSTM EHR model[30]: is a deep, autoregressive LSTM model, adapted to generate structured patient records rather than unstructured text as it had previously been utilized, which is directly analogous to the HALO-coarse model but uses LSTM blocks instead of transformer decoder blocks.

- SynTEG[15]: is a GAN-based model that uses a transformer and LSTM-based encoder model to generate embeddings of EHRs up

to a given visit before feeding those embeddings into a conditional GAN which generates the next visit.

- EVA[16]: is a VAE-based model that uses a bidirectional-LSTM encoder and CNN-based decoder (using deconvolutions to expand the latent encoding to the proper temporal dimension and then masked, diluted 1D convolutions to build the records in an autoregressive manner). The only change we made was to convert the output from one-hot code combinations to multi-hot code probabilities to allow for greater representative power.

## Evaluating EHR language modeling

The first evaluation is conducted by predicting the probabilities and outputs of the test set. In this phase, we assess the performance of HALO against two multi-hot language model baselines, namely HALO-Coarse and LSTM. These baselines explicitly generate a probability distribution without accessing the entire input. It's worth noting that other baseline models, such as the GAN-based SynTEG model, the VAE-based EVA model, and the GPT model, cannot be directly compared in this task because those methods do not make a single probability prediction for each code within the visit.

Our first evaluation aims to assess the capability of the models to predict the presence of potential medical codes, given a patient's past medical history and the previous codes from the current visit. Note that we explore different orderings of codes (such as most to least prevalent, alphanumeric, random, etc.) but find no noticeable differences, displaying the results of such an exploration in our supplementary information and settling on a random ordering throughout our experiments. This evaluation is crucial in showcasing a model's ability to learn patterns from the patient population and its potential to perform well in various patient simulation and extension applications. We show the results in Table 1 where we see that HALO outperforms the two compared language model architectures. Upon closer examination, we observed that the LSTM baseline model struggled with the complexity and size of the outpatient EHR dataset, while our proposed model HALO performed comparably to the HALO-Coarse ablation baseline. In contrast, in the inpatient EHR setting, where the visits are shorter but contain more codes, HALO's multi-granularity approach proved to be highly effective. Specifically, the model achieved a notable 4% reduction in binary cross-entropy (BCE) loss and a 17% increase in F1 Score on test data when compared to the single granularity HALO-Coarse model. Notably, both HALO models significantly outperformed the LSTM baseline in this setting. These results highlight the significant value of our multi-granularity approach in handling the complex and diverse nature of medical codes in different EHR settings.

Additionally, we present perplexity, which evaluates the probability or likelihood of the test set as quantified by a model trained on the training set, normalized by the unit of consideration that we are interested in. In our case, this normalizing unit is the number of medical codes in a patient's medical record (or equivalently number of ones in **R**). Perplexity is a metric found commonly in the wider generative modeling domain, especially on the task of natural language generation (e.g. ref. 29). We introduce it to the task of synthetic EHR

## Table 1 | Test set modeling metrics

| | Outpatient EHR | | | Inpatient EHR | | |
|---|---|---|---|---|---|---|
| | BCE loss | F1 score | PP per code | BCE loss | F1 score | PP per code |
| LSTM | $7.744 \times 10^{-4}$ | 0 | 660.204 | $2.600 \times 10^{-4}$ | 0.193 | 74.565 |
| HALO-Coarse | $1.631 \times 10^{-4}$ | **0.829** | 3.927 | $2.019 \times 10^{-4}$ | 0.343 | 28.448 |
| HALO | $\mathbf{1.624 \times 10^{-4}}$ | 0.828 | **3.903** | $\mathbf{1.932 \times 10^{-4}}$ | **0.414** | **24.664** |

We include each of our autoregressive, predictive, and likelihood-based models. The bold value denotes the best results. Baseline methods SynTEG, EVA, and GPT are all omitted here because they either do not produce a probability distribution, peek at the outputs, or utilize a different, non-comparable data representation. HALO outperforms both of the baselines, achieving up to a 4% decrease in testset BCE loss, a 17% increase in F1 score, and a 13% lower perplexity per present code as compared to the leading HALO-Coarse baseline. Source data are provided as a Source Data file.

generation here. Perplexity is defined mathematically by

$$PP(D) = \sqrt[N]{\frac{1}{P(D)}}$$

$$= \sqrt[N]{\frac{1}{P(\mathbf{R}^{(1)}, \cdots, \mathbf{R}^{(|D|)})}} \qquad (2)$$

$$= \sqrt[N]{\frac{1}{P(\mathbf{R}^{(1)}) \cdots P(\mathbf{R}^{(|D|)})}}$$

where $D$ is the test dataset and $\mathbf{R}^{(t)}$ is the $t$th record in $D$. In practice we calculate the values by summing their log probabilities, using the equivalent form

$$PP(D) = \exp\left(-\frac{1}{N}\sum_{\mathbf{R} \in D} \log P(\mathbf{R})\right) \qquad (3)$$

The normalized value then also corresponds to how many of the different normalizing units (medical codes) one would have to randomly pick between on average to achieve the same probability. The results of the perplexity evaluation are shown in Table 1 as well. We see similar results as with the classification evaluation with both HALO and HALO-Coarse performing very well on the outpatient EHR dataset (with HALO performing slightly better) as the LSTM baseline struggles, and HALO easily outpacing both baseline methods in this likelihood evaluation for the inpatient EHR dataset, producing a 13% lower perplexity per present code as compared to the HALO-Coarse architecture without the inter-visit modeling. Thus, in both of these test set evaluations, we see that HALO is much more effective in terms of modeling the underlying distribution of EHRs.

## Statistical similarity to real EHRs

The second analysis evaluates the statistical similarity of the generated and real data. For each method, we generate a synthetic dataset of the same size as the training dataset. We then compare the unigram and bigram (both within the same visit and across consecutive visits) probabilities for each unique code and pair of codes within the real and synthetic datasets.

**Statistical comparison results.** We evaluate the data at the visit and record level, considering approximately 10,000 individual codes and over a million bigram codes. We also compare various aggregate statistics, such as the number of visits per record, medical codes per visit, and prevalence of chronic disease labels. The code probability results are presented in Fig. 3, and the aggregate statistics are in Table 2.

Additionally, we provide $R^2$ values for visit-level normalized code probabilities in our high-dimensional outpatient EHR dataset and a lower-dimensional setting. The details can be found in Table 3.

Furthermore, an interactive visualization of 1000 randomly selected code-level disease prevalence comparisons between our method and real data is accessible at https://vega.github.io/. It allows zooming, panning, and hovering over points for specific disease names. Finally, we provide chronic disease label probabilities, full visit level code probability plots, probability densities underlying the aggregated statistics, and a discussion of the various failure modes of our baseline methods for that evaluation in our supplementary information. HALO again outperforms the baseline methods in each evaluation.

**Key findings.** We observe that besides the GPT baseline struggling with the complexity of the outpatient EHR dataset in terms of stopping the record generation (as is common to many language models in the text generation domain as their overall quality decays for long sequences, and the lack of visit level grouping in its data representation causes its sequences to be considerably longer), the language model architectures (GPT, LSTM, HALO-Coarse, and HALO) can model both the shape of the synthetic records and the temporal dependencies much better on average than the VAE and especially GAN-based baselines. While each of the compared methods models the unigram code probabilities relatively well, the temporal modeling is better shown in the overall synthetic record and visit lengths, the generation of chronic disease labels, and the sequential bigram evaluation. SynTEG, EVA, and the LSTM baseline thus struggle with these evaluations (with the LSTM baseline struggling largely due simply to overall weakness).

The LSTM and HALO-Coarse language model baselines then falter with respect to same-visit bigram probabilities due to their lack of intra-visit dependency modeling while the GPT baseline which models each code individually and so offers that intra-visit modeling can maintain relatively stronger performance there. HALO can combine and build on each baseline's strengths without any weaknesses, using the compact multi-hot representation to offer a powerful model that does not struggle with any length or feature of data while simultaneously maintaining the intra-visit modeling in an even more powerful and structured way. As such, it can best maintain performance in this high-dimensional setting and produces state-of-the-art results that closely model the true training data in all settings from record and visit lengths, label probabilities, and all combinations of code probabilities. This signifies that HALO is capable of generating data that looks realistic.

## Accurate disease phenotyping using synthetic EHRs

The final evaluation explores the utility of the synthetic datasets for training disease classifiers. To this end, we utilize two different synthetically supplemented data setups and machine learning classifiers to predict chronic disease labels based on patients' visits. In each of the two setups, we use a simple bidirectional LSTM with a single-layer fully connected head classifier to predict chronic disease label(s) based on a patient's visits.

**Accurate disease phenotyping.** In the first data setup, we assess model performance in real-world scenarios using synthetic training data. We conduct experiments for each of the 11 chronic disease labels in the outpatient EHR dataset, sourced from the Centers for Medicare and Medicaid Services and the SynPUF dataset[31]. Additionally, we perform experiments for each of the 25 chronic diseases in the inpatient EHR dataset, based on the benchmark proposed in ref. 27.

For each chronic disease, we randomly extract 2500 records for training, ensuring a balanced distribution of positive and negative labels (50–50). This process is repeated across our six synthetic datasets (one for each method) and one real training dataset, resulting in a total of seven balanced training datasets. The selected size of 2500 records strikes a balance between having enough training data for machine-learning models and maintaining sufficient positive labels for each disease.

We train classifiers on each of these datasets and select the best model for each dataset using a validation set of 250 records, equally representing both classes. Finally, we evaluate the models on test sets consisting of 500 records, equally representing both classes, from the original test set comprising real patient data.

We display the average accuracy and F1 score for each synthetic dataset from each of the compared models as well as the real training data across each of the chronic disease labels in Table 4. Note that we provide the standard deviations of each metric in either table as well, but most of that deviation stems from differences between tasks rather than inconsistent performance within each model.

We provide a full set of results by chronic disease label and also additional synthetically augmented outpatient results in our supplementary information. In both datasets, we can see that each synthetic

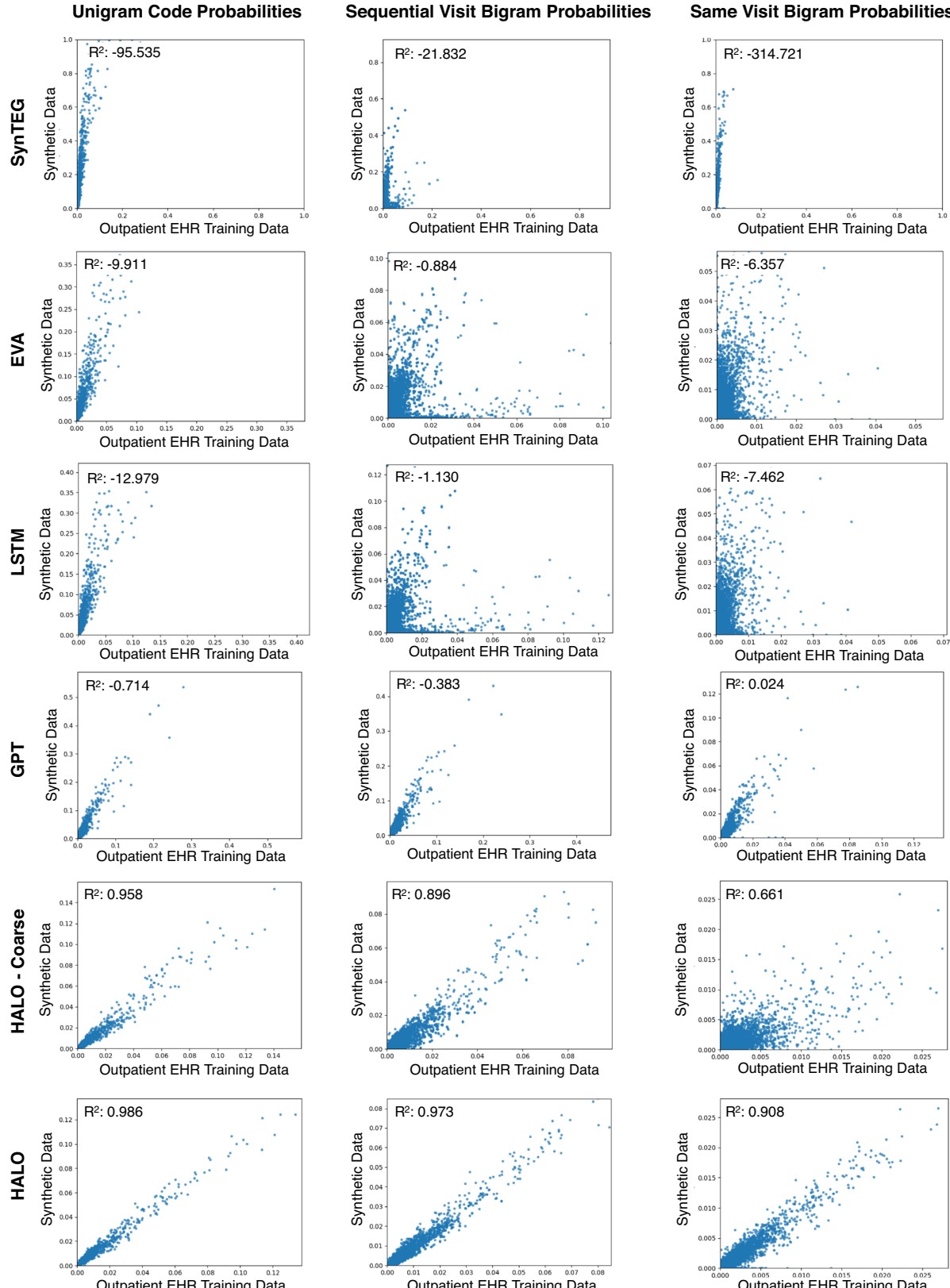

**Fig. 3 | Code probability plots.** These plots show the Unigram, Sequential Visit Bigram, and Same Record Bigram probabilities for each synthetic dataset. With the exception of SynTEG, all models exhibit some correlation in the unigram and temporal bigram evaluations, but many have weak correlations or consistently yield higher synthetic probabilities due to a lack of temporal consistency and repetition across visits in the records. HALO and to a lesser extent, HALO-Coarse perform the best in all settings, while HALO is the only one that can realistically produce pairs of codes within and across visits and achieve state-of-the-art results.

data of GPT, `HALO`-Coarse, and `HALO` largely maintains the performance of real data and offers large improvements over the SynTEG, EVA, and LSTM baselines. `HALO`'s synthetic data offers the best prediction results.

**Phenotyping of rare conditions.** We evaluate the utility of synthetic EHR data in identifying uncommon conditions. We created a highly imbalanced dataset of patients labeled with cancer chronic disease from the outpatient EHR dataset. The dataset comprised 50,000 EHR records without the cancer chronic disease label and only 1000 records with the label.

Using this imbalanced data, we trained a classifier and compared its performance to classifiers trained on balanced datasets. For balancing, we added 49,000 positively labeled synthetic records and also used another classifier trained on a dataset balanced using real records.

The evaluation results are summarized in Table 5. Notably, `HALO` outperformed all baselines, exhibiting significant improvements on the original unbalanced dataset as well as the synthetically augmented datasets. It approached the upper bound performance of the ideal balanced dataset.

This experiment underscores the potential of synthetic EHR data in supporting the identification of uncommon conditions.

### Realistic continuous variables in synthetic EHRs

We conclude with a brief exploration to demonstrate the viability of our discretized representation of continuous values, and `HALO`'s effectiveness in using it to model those variables. We build new training datasets including visit gaps in the outpatient EHR dataset and lab values in the inpatient EHR dataset. We use these datasets to train a new version of our model and generate another synthetic dataset of 250,000 and 45,000 records, respectively.

**Table 4 | Chronic disease classification model performance trained on synthetic data**

|  | Outpatient EHR | | Inpatient EHR | |
| --- | --- | --- | --- | --- |
|  | Avg. accuracy | Avg. F1 score | Avg. accuracy | Avg. F1 score |
| EVA | 0.508 ± 0.02 | 0.283 ± 0.26 | 0.5356 ± 0.05 | 0.580 ± 0.05 |
| SynTEG | 0.507 ± 0.03 | 0.514 ± 0.20 | 0.539 ± 0.06 | 0.438 ± 0.06 |
| LSTM | 0.506 ± 0.02 | 0.467 ± 0.28 | 0.522 ± 0.04 | 0.565 ± 0.04 |
| GPT | 0.851 ± 0.03 | 0.854 ± 0.03 | **0.877 ± 0.05** | **0.881 ± 0.05** |
| HALO-Coarse | 0.867 ± 0.03 | 0.863 ± 0.03 | 0.863 ± 0.05 | 0.865 ± 0.05 |
| HALO | **0.879 ± 0.03** | **0.878 ± 0.03** | **0.882 ± 0.04** | **0.884 ± 0.04** |
| Real data | 0.891 ± 0.03 | 0.895 ± 0.03 | 0.938 ± 0.04 | 0.937 ± 0.04 |

We compared the average performance in terms of accuracy and F1 Score for each of the 11 chronic disease labels in our outpatient dataset and 25 chronic disease labels in our inpatient dataset. The models were trained on each of our synthetic datasets and tested on real data. The reported values represent the mean and standard deviation across the tasks, with bold values indicating the best results. GPT, HALO-Coarse, and HALO's data offer large improvements over the other baselines and perform similarly to real training data. HALO's synthetic data performs the best with the highest average performance of all synthetic methods. Source data are provided as a Source Data file.

**Table 5 | Rare disease detection performance on synthetic balanced datasets**

|  | BCE loss | Accuracy | F1 score | AUROC |
| --- | --- | --- | --- | --- |
| Original imbalanced | 0.693 | 0.497 | 0.013 | 0.417 |
| Balanced with real data | 0.127 | 0.951 | 0.951 | 0.989 |
| EVA | 0.615 | 0.695 | 0.705 | 0.730 |
| SynTEG | 0.598 | 0.735 | 0.758 | 0.786 |
| LSTM | 0.593 | 0.702 | 0.714 | 0.743 |
| GPT | 0.472 | 0.880 | 0.869 | 0.956 |
| HALO-Coarse | 0.265 | 0.918 | 0.916 | 0.959 |
| HALO | **0.192** | **0.931** | **0.931** | **0.976** |

We present the classification results on the test set for the simulated rare-disease detection task. We compare models trained on datasets balanced using each synthetic dataset against models trained on the original imbalanced data (representing the rare disease dataset). Additionally, we compare the results against an upper-bound ideal dataset balanced using real data. The best results are highlighted in bold. Among the evaluated models, EVA and SynTEG exhibit limited utility, while the language model architectures LSTM, GPT, and HALO-Coarse offer substantial value. HALO achieves state-of-the-art performance, closely approaching the results of a true, balanced dataset. The source data can be found in the provided Source Data file.

**Table 2 | Aggregate statistics regarding the shape of training and compared synthetic datasets**

|  | Outpatient EHR | | Inpatient EHR | |
| --- | --- | --- | --- | --- |
|  | Record length mean (std. dev.) | Visit length mean (std. dev.) | Record length mean (std. dev.) | Visit length mean (std. dev.) |
| EVA | 29.49 (28.88) | 3.35 (1.71) | 1.20 (0.723) | 11.92 (3.665) |
| SynTEG | 93.00 (2.30) | 3.70 (4.10) | 27.55 (3.34) | 5.93 (10.96) |
| LSTM | 32.04 (27.14) | 3.22 (1.64) | 1.30 (0.56) | 9.53 (2.91) |
| GPT | 95.72 (3.37) | 2.70 (1.73) | 1.26 (0.73) | 9.67 (5.45) |
| HALO-Coarse | 35.26 (31.87) | 3.77 (2.23) | 1.13 (0.39) | 11.21 (3.91) |
| HALO | 36.19 (33.41) | 3.93 (2.72) | 1.31 (0.84) | 11.93 (6.45) |
| Train data | 34.18 (32.35) | 3.52 (2.18) | 1.27 (0.92) | 11.68 (5.70) |

Aggregate statistics on the number of visits per record and the number of codes per visit. The values are mean (std). HALO outperformed all the baselines while closely approximating the distribution of the true training data. Source data are provided as a Source Data file.

**Table 3 | Code probability correlations $R^2$ between training and synthetic datasets**

|  | High-dimensional outpatient EHR | | | Low-dimensional outpatient EHR | | |
| --- | --- | --- | --- | --- | --- | --- |
|  | Unigram code probabilities | Sequential visit bigram probabilities | Same visit bigram probabilities | Unigram code probabilities | Sequential visit bigram probabilities | Same visit bigram probabilities |
| EVA | 0.910 | 0.082 | 0.128 | 0.957 | 0.134 | 0.225 |
| SynTEG | **0.915** | 0.355 | 0.082 | 0.784 | 0.315 | 0.211 |
| LSTM | 0.900 | 0.077 | 0.127 | **0.962** | 0.135 | 0.225 |
| GPT | 0.743 | 0.382 | 0.262 | 0.924 | 0.626 | 0.515 |
| HALO-Coarse | 0.794 | 0.357 | 0.176 | 0.882 | 0.503 | 0.247 |
| HALO | 0.914 | **0.508** | **0.362** | 0.949 | **0.686** | **0.562** |

The values are $R^2$ values to measure the correlations of the three types of code probabilities for different synthetic datasets against the training data in both high-dimensional and low-dimensional settings. Bold values denote the best results. Although the results showed a drop in performance for each method in the high-dimensional setting, HALO was able to maintain strong performance with minimal decline. Overall, our proposed method achieved state-of-the-art performance, outperforming the baselines in both bigram evaluations in low and high-dimensional settings. Source data are provided as a Source Data file.

We then show that the distributions of those variables match the real values. In Fig. 4a and b, we show that HALO accurately replicates the distribution of gaps between patient visits and the pattern of shorter gaps for longer records, respectively. These captured nuanced patterns are on top of the aggregate mean gaps being very similar as well. There are 33.53 days between visits on average within the real outpatient EHR data and 35.77 days on average for HALO's synthetic data.

Using the inpatient dataset, we then demonstrate that HALO replicates not only the presence (in Fig. 4c) but also the average values (in Fig. 4d) of performed lab tests. Specific labs included (corresponding to points in those two plots) are included in our supplementary information. Overall, HALO's approach to continuous variables is effective, and it has the potential to generate comprehensive synthetic patient records with multiple variables of different types.

### Privacy evaluation of synthetic EHRs

In addition to demonstrating the high fidelity of synthetic EHRs generated by HALO, we want to ensure that the privacy of the patients within the original training dataset is protected. To that end, we conduct a commonly used membership inference attack to test its

identification risk, and we provide the results of two more evaluations in our supplementary information.

**Membership inference attack.** The evaluation is the ability to thwart a membership inference attack. These attacks aim to determine whether any specific real patient record was used in the training dataset to generate the synthetic records. Membership inference attacks are a well-known privacy test in the field of synthetic EHR generation, and addressing them is crucial to ensure the privacy and confidentiality of patient identities.

To demonstrate that HALO is not susceptible to such an attack, we show that we can prevent two different attempts at a membership inference attack based on the synthetic data generator and the synthetic dataset itself. We generate an attack dataset by first selecting 100,000 records from each real dataset used for training and assigning them a positive label. Then we select 100,000 records from the remaining records not used for training as the negative label set.

Next, we conduct two attacks:
- In the Model Attack, we label the 100,000 records with the highest log probability from the model as positive, predicting that they were part of the training dataset.

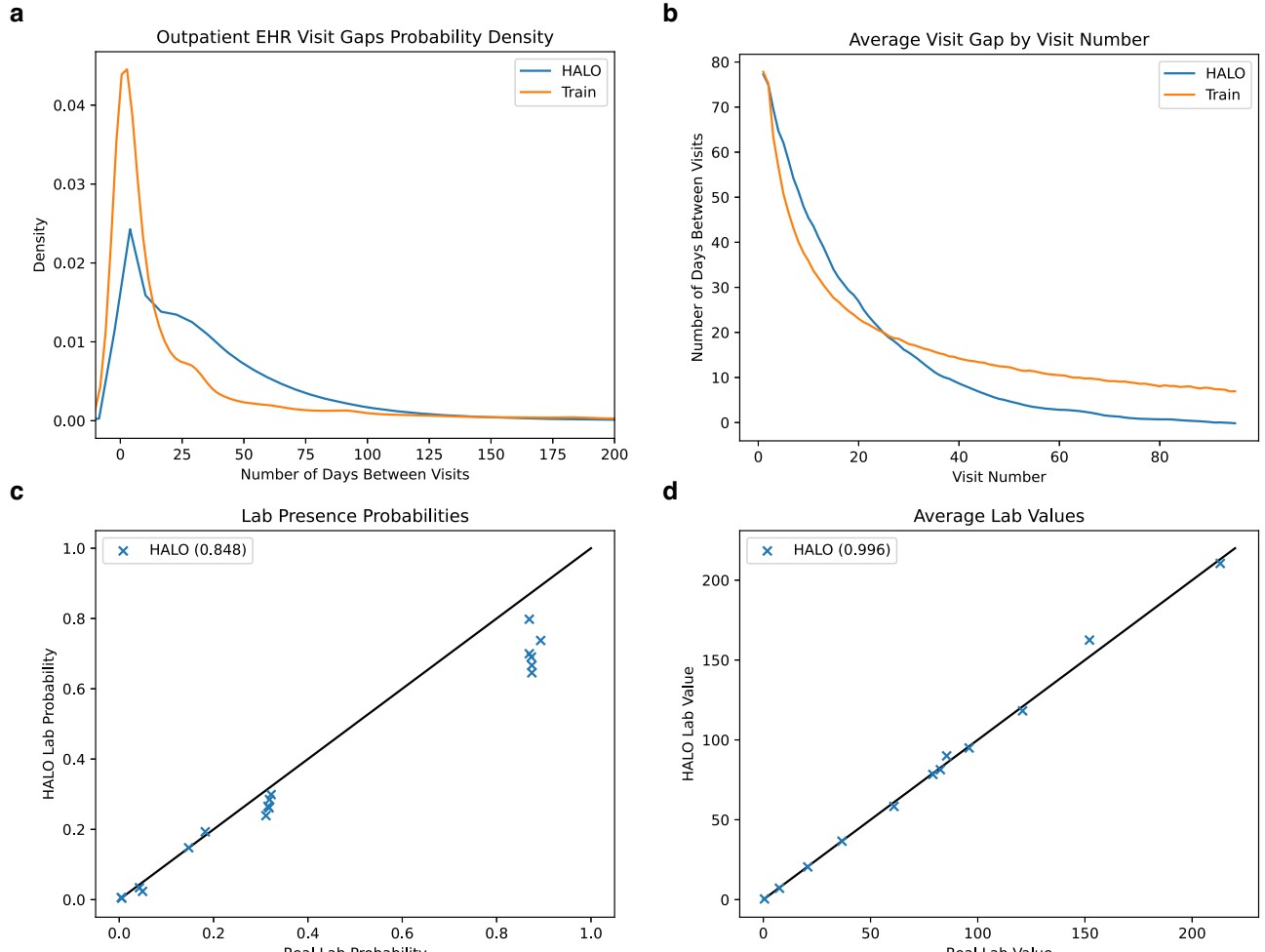

**Fig. 4 | Continuous variable generation performance:** HALO **effectively captures the distribution of continuous variables through its discretization approach, as demonstrated in four scenarios. a** Inter-visit gap probability density: The probability density of inter-visit gaps indicates that HALO closely approximates the true shape of real data. **b** Inter-visit gap by visit number: The mean visit gap, as per visit number, across both real and synthetic datasets reveals that HALO accurately captures the pattern of patients with many records, showing shorter gaps in their subsequent visits. **c** Lab presence probabilities: The probability of binary lab presence demonstrates that HALO accurately generates lab variables, even when discretized across multiple variables. **d** Mean lab values: The average value of labs, when present, confirms that HALO's synthetic labs closely resemble those of the real dataset. Values in parentheses are $R^2$.

**Table 6 | Membership inference attack results**

| | Outpatient EHR | | | Inpatient EHR | | |
|---|---|---|---|---|---|---|
| | Acc. | Precision | Recall | Acc. | Precision | Recall |
| HALO Dataset Attack | 0.501 | 0.501 | 0.501 | 0.492 | 0.491 | 0.477 |
| HALO Model Attack | 0.509 | 0.509 | 0.509 | 0.515 | 0.515 | 0.515 |
| EVA Dataset Attack | 0.498 | 0.498 | 0.496 | 0.493 | 0.493 | 0.477 |
| SynTEG Dataset Attack | 0.500 | 0.500 | 0.500 | 0.491 | 0.491 | 0.467 |
| LSTM Dataset Attack | 0.499 | 0.499 | 0.496 | 0.494 | 0.494 | 0.481 |
| GPT Dataset Attack | 0.500 | 0.500 | 0.500 | 0.492 | 0.491 | 0.455 |
| HALO-Coarse Dataset Attack | 0.500 | 0.500 | 0.499 | 0.491 | 0.491 | 0.462 |

For each record in the attack dataset, we find both the log probability of the record from the trained model (Model Attack) and the hamming distance to the closest record in the synthetic dataset (Dataset Attack). The attacks then label half of the records with the highest probability or lowest distance records, respectively, as in the training set. We see that the accuracy for either attack is right around 50%, which is similar to a random guess. This indicates that the synthetic dataset and the model do not reveal any patient-identifying information about the original training datasets. We also find that each baseline synthetic dataset similarly thwarts the dataset attack. Source data are provided as a Source Data file.

- In the Dataset Attack, we label the 100,000 records with the lowest hamming distance to the closest record in the synthetic dataset as positive. We pick a hamming distance (equivalent to Manhattan Distance in our binary setting) as our distance metric between patient records throughout our privacy evaluations in accordance with Yan et al.[32], but any distance metric could be substituted interchangeably. These two attacks allow us to test the ability of the synthetic dataset to prevent an attacker from inferring whether a real record was used in the training dataset.

We show the results of the classifications from the attacks in Table 6. The accuracy of both attacks on both datasets is ~50%, which is similar to a random guess. This shows that neither the model nor the synthetic dataset reveals any meaningful or compromising information about the patient identity in the training dataset. We also perform the dataset attack with each of our baseline datasets and see that each similarly accomplishes it, achieving a similar probability at around 50%. Note that we do not perform the model attack with the baseline models because most of them cannot offer a probability output of input patient records, and the dataset-based attack is the standard one used throughout literature in this domain.

Beyond the membership inference attack, we also show that HALO passes attribute inference attack and nearest neighbor adversarial accuracy[33] evaluations in our supplementary information.

## Discussion

In this paper, we proposed a method HALO for generating high-dimensional synthetic longitudinal EHR data. Our method is specifically designed to handle the sequential, multi-granular, and high-dimensional nature of electronic health records by generating an explicit probability distribution over the codes, visits, and records, and HALO can generate realistic data without needing to aggregate or remove any codes as past approaches have unanimously done. We then showed that HALO can produce incredibly realistic synthetic EHR data. Specifically, we showed that HALO can capture the probability distribution underlying the records better than other language model baselines and then produce a synthetic dataset that both looks similar to and offers the utility of real patient records as measured by medical code occurrence probabilities and machine learning classification tasks augmented with synthetic data. Finally, we also show that our method offers this performance without compromising privacy through several privacy evaluations.

In conclusion, one of the key advantages of HALO is its ability to generate binary sequences that are over a million variables in length. Its impressive performance makes it a promising avenue for developing and sharing realistic but synthetic EHR datasets that can support diverse applications. This represents an exciting opportunity to expand the use of synthetic data in the healthcare field and could help

address some of the challenges associated with data privacy and security.

While we have shown the impressive performance of HALO in both producing high-quality, high-fidelity, and privacy-preserving, we now briefly discuss some remaining limitations. First, the architecture is designed in the model of a large language model. While the multi-modal setup allows the model to condition on more patterns per data point and learn more efficiently, our high-performing generator still requires relatively large training datasets which might not be available in some settings.

Another important aspect of our model is that it generates synthetic records through a probabilistic process. While it learns real-world patterns during training, there is still a chance that some generated records may not be clinically meaningful. However, this risk can be mitigated through postprocessing with clinical rules that validate the synthetic records. If our model is deployed in the real world, it is important to consider implementing such postprocessing steps to ensure that only clinically relevant synthetic records are produced.

Finally, our HALO model focuses on generating longitudinal EHR data, such as medical codes and lab results. However, other crucial data modalities, such as clinical notes and medical images, are not yet covered by the model. To generate fully comprehensive patient records that include all modalities, it will be necessary to use diverse training data and develop multiple models to handle each modality. This exciting avenue of research is a promising future direction.

## Methods

Our study has acquired exempt status from Institutional Review Board (IRB) approval. This study has been found to be exempt pursuant to 45CFR46.104(d)(4) "Secondary research for which consent is not required: Secondary research uses of identifiable private information, if (i) The identifiable private information is publicly available; AND (ii) Information is recorded by the investigator in such a manner that the identity of the human subjects cannot readily be ascertained directly or through identifiers linked to the subjects, the investigator does not contact the subjects, and the investigator will not re-identify subjects."

### Background and related work

Of all the EHR generation methods, rule-based approaches, such as Synthea[34] or SynPUF[31], have proven to be the most effective in delivering practical value. These simple approaches either offer de-identification of real records by combining data across multiple patients in a sufficiently privacy-preserving way[31], simulation of patients within a complex yet constrained rule-based system[34], Bayesian probabilistic modeling of aggregated, non-temporal patient records[35], or proprietary method without detailed explanation[36–38]. Many of these systems can only produce synthetic patient data with

limited capacity in realism and utility. We focus instead on ML methods that have the potential to generate realistic high-dimensional synthetic patient data.

**GAN-based methods.** Many synthetic data generation methods use generative adversarial networks (GANs), which involve a generator that creates realistic data, and a discriminator that decides if the data is real or fake[39]. The GANs have been applied to patient record generation first in ref. 19 followed by many other GAN-based approaches[15,17,18,20–24,40]. However, GANs have limitations when generating sequential data like EHRs. They usually only produce one output (no time connections) and so most EHR generation methods aggregate EHR data into one time step[22–24], create a representation of EHR data[18], or do both[19,20].

GANs also struggle with high dimensional and sparse data like real-world EHR, limiting all existing synthetic EHR GAN approaches to produce relatively low dimensional data through the aggregation of visits and medical codes or removal of rare codes. For example, there are a few methods in this category which do generate longitudinal data. LongGAN[40] and EHR-M-GAN[21] both focus only on dense lab time series of under a hundred dimensions. CorGAN[17] generates records with 1071 distinct codes, and the current state-of-the-art GAN approach that we baseline against, SynTEG[15], both combines and removes rare codes before arriving at a final dimensionality of 1276.

While GANs have the potential to be conditioned on external factors and labels, such as demographics or disease phenotype labels, the ability to do so has not been extensively explored in existing works on EHR generation. Moreover, there are only a limited number of approaches that can generate synthetic EHR data tailored to specific diseases. For example, SmoothGAN[24] focuses on aggregated lab and medication information and does not model individual visits; EHR-M-GAN[21] offers conditional and sequential capabilities but for low dimensional (under 100 dimensions) lab time-series information; CONAN and MaskEHR[18,41] model only a single rare-disease population for data augmentation; and EMR-WGAN and HGAN[22,23] can only model low-dimensional (both under 1000 dimensions) aggregated EHRs.

**Deep sequential methods.** Accurately modeling the longitudinal nature of EHRs is crucial for realistic EHR generation. In recent years, two methods have shown progress in generating sequential EHRs by using either a GAN or a VAE to condition representations of past patient visits to generate current visits[15,16]. Specifically, SynTEG[15] models the time between visits, and EVA[16] offers a conditional variant. In our experiments, we compare HALO to these two models. However, both SynTEG and EVA often need to perform preprocessing steps to reduce the dimensionality of the vocabulary by aggregating medical codes and removing rare codes.

**Language models.** Our objective is to develop an improved method for generating realistic and high-dimensional EHR data by drawing inspiration from natural language generation. Language generation models predict the next word based on the preceding words, thereby learning a probability distribution of languages. Similarly, EHR models predict the next visit based on past visits. Also our proposed method provides an explicit probability output that allows for direct modeling and evaluation of the underlying data distribution. This approach is particularly beneficial in accurately capturing the complex and high-dimensional nature of EHR data.

The Transformer architecture, introduced in ref. 42, has revolutionized natural language processing and enabled the development of large, attention-based models like BERT[43] and GPT[26,29,44]. Among these models, we draw inspiration from GPT, which relies on a stack of Transformer decoder blocks that use masking to predict the next set of probabilities in parallel, allowing for fast training and scalability. However, applying language models directly to EHR data poses unique challenges. Unlike natural language sequences, EHR data exhibits a hierarchical structure that must be captured, with medical codes associated with specific patient visits, and visits associated with individual patients. Additionally, EHR data contains heterogeneous elements, including demographic variables, structured medical codes, and numeric lab measures, not all of which are discrete tokens. Addressing these challenges requires approaches that leverage the strengths of language models while adapting them to the peculiarities of EHR data.

## Hierarchical autoregressive language model (HALO)

We model the probability of patient record $\mathbf{R}$, $P(\mathbf{R})$, via a hierarchical autoregressive model, which utilizes both visit- and code-level structures of a patient record. First, it factorizes the probability along the visit level using the autoregressive identity by

$$
\begin{aligned}
P(\mathbf{R}) &= P(\mathbf{v}_s, \mathbf{v}_l, \cdots, \mathbf{v}_T, \mathbf{v}_e) \\
&= P(\mathbf{v}_s) P(\mathbf{v}_l | \mathbf{v}_s) P(\mathbf{v}_1 | \mathbf{v}_s, \mathbf{v}_l) \cdots P(\mathbf{v}_e | \mathbf{v}_s, \mathbf{v}_l, \cdots, \mathbf{v}_T)
\end{aligned}
\tag{4}
$$

to produce what we call our coarse autoregressive sequence. We then continue to factorize the probability of visits further along the code level by converting

$$
\begin{aligned}
P(\mathbf{v}_t | \mathbf{v}_s, \cdots, \mathbf{v}_{t-1}) &= P(c_t^1 | \mathbf{v}_s, \cdots, \mathbf{v}_{t-1}) P(c_t^2 | \mathbf{v}_s, \cdots, \mathbf{v}_{t-1}, c_t^1) \\
&\quad \cdots P(c_t^C | \mathbf{v}_s, \cdots, \mathbf{v}_{t-1}, c_t^1, \cdots, c_t^{C-1})
\end{aligned}
\tag{5}
$$

into what we call our fine autoregressive sequence. This final probability is then rewritten as the product

$$
P(\mathbf{R}) = \prod_t^C \prod_i^C P(c_t^i | \mathbf{v}_s, \cdots, \mathbf{v}_{t-1}, c_t^1, \cdots, c_t^{i-1})
\tag{6}
$$

where the probability of each code is based on each of the previous visits and each of the previous codes in the current visit. Our multi-granularity approach enables the modeling of high-dimensional sequences of many binary variables per record. This is achieved by grouping prior information into significantly fewer multivariate time steps for previous visits while retaining the full autoregressive modeling capability for each current visit. Our HALO architecture is designed to reflect this powerful yet compact model, with a powerful and efficient structure divided into two distinct granularity levels: visit level and code level. This allows for each code to be conditioned on all previous visits and the past codes of the current visit.

**Visit-level module.** We begin with the coarse, visit-level granularity. We use a stack of $M$ transformer decoder blocks, which have shown to be effective in the high-dimensional domain of natural language processing, to generate a sequence of visit-level histories, where the $t$-th element in the sequence, $\mathbf{h}_t^{(M)} \in \mathbb{R}^{n_{emb}}$, is an embedding that represents all of a patient's medical history through their $t$-th visit. Those histories then combine to form $\mathbf{H}^{(M)} \in \mathbb{R}^{(T+3) \times n_{emb}}$ (where the 3 in $T+3$ includes the start, label, and end visits), the output of the first module which serves the purpose of the $\mathbf{v}_s, \mathbf{v}_l, \mathbf{v}_1, \cdots \mathbf{v}_{t-1}$ priors in Eq. (6).

To encode each of the multi-hot visit representations $[\mathbf{v}_1 \cdots \mathbf{v}_n]$ into a fixed-length vector in $\mathbb{R}^{n_{emb}}$, we employ an embedding layer that includes two trainable parameter matrices: a code embedding matrix $\mathbf{W}_c$ and a positional embedding matrix $\mathbf{W}_p$. The code embedding matrix maps each visit code to a dense vector representation, while the positional embedding matrix captures the relative position of each visit in the sequence. Next, we use a decoder model consisting of $M = 12$ transformer decoder blocks to generate a series of visit history representations, which summarize the information contained in all previous visits in the coarse, visit-level sequence. The transformer

decoder blocks employ masked multi-head self-attention, which allows the model to attend to all previous visits while preventing information leakage from future visits. This process is written more formally as

$$\mathbf{H}^{(0)} = \mathbf{R}\mathbf{W}_e + \mathbf{W}_p$$
$$\mathbf{H}^{(m)} = \text{transformer\_block}(\mathbf{H}^{(m-1)}) \quad \forall m \in [1, M] \tag{7}$$

where $\mathbf{R} \in \mathbb{R}^{(T+3) \times C}$ is the patient record matrix representation, $\mathbf{W}_e \in \mathbb{R}^{C \times n_{emb}}$ is the code embedding matrix, $\mathbf{W}_p \in \mathbb{R}^{(T+2) \times n_{emb}}$ is the positional embedding matrix (to recapture the position and order of the sequence of visits), and each transformer block is based on a decoder block from the original transformer architecture[42] which we describe in more detail in our supplementary information.

Thus, having processed the multi-hot patient visits through the initial, coarse visit-level module of our architecture, we obtain a sequence of visit history representations $\mathbf{H}^{(M)}$, which capture the collective information of all previous visits up to each time step. These representations provide a compressed summary of the patient's visit history, enabling downstream modules to make predictions based on the patient's medical trajectory.

**Code-level module.** However, we still need to add in the code-level priors and generate output probabilities. To construct the input for the fine, code-level module, we offset and concatenate the previous module's visit history embedding outputs with the original record input, $\mathbf{R}$. Specifically, we append the first $T+2$ visit histories with the last $T+2$ visit representations $[\mathbf{v}_l, \mathbf{v}_1, \cdots, \mathbf{v}_T, \mathbf{v}_e]$ to create $\mathbf{H}'^{(0)}$. Each of the $T+2$ inputs in $\mathbf{H}'^{(0)}$ has a representation of the history of all the previous visits and the codes of the current visit, mirroring both the visit and code priors in Equation (6). The final input representation $\mathbf{H}'^{(0)}$ has size $\mathbb{R}^{(T+2) \times (n_{emb} + C)}$.

To model the distribution of each $P(c_t^i)$, this $\mathbf{H}'^{(0)}$ is then fed through $N = 2$ masked linear layers which maintain the same dimensionality and use upper triangular masking of the weight matrix to ensure that they preserve the autoregressive property of the probabilities (and have a ReLU activation function between layers). These linear layers are able to efficiently model the high-dimensional, intra-visit patterns where other sequential approaches such as additional recurrent or transformer modules would run out of memory. The probabilities are generated formally by

$$\mathbf{H}'^{(0)} = \text{offset\_and\_concat}(\mathbf{H}^{(M)}, \mathbf{R})$$
$$\mathbf{H}'^{(n)} = \text{masked\_linear}(\mathbf{H}'^{(n-1)}) \quad \forall n \in [1, N] \tag{8}$$
$$\mathbf{O} = \text{sigmoid}(\mathbf{H}'^{(N)}[:, n_{emb}:])$$

where the submatrix indexing at the end removes the visit-level history embedding portions of each vector to extract just the code probabilities, and the masked linear layers are achieved by

$$\mathbf{H}'^{(n)} = \max(0, \mathbf{H}'^{(n-1)}(\mathbf{W}^{(n)} \odot \mathbf{M}) + \mathbf{b}^{(n)}) \tag{9}$$

where the max function is omitted for the final fine layer (sigmoid is used instead), $\odot$ is element-wise matrix multiplication, $\mathbf{M} \in \mathbb{R}^{(n_{emb} + C) \times (n_{emb} + C)}$ is the upper triangular masking matrix (with ones in the upper triangular portion and zeros in the lower portion) to preserve the autoregressive property, and $\mathbf{W}^{(n)} \in \mathbb{R}^{(n_{emb} + C) \times (n_{emb} + C)}$ and $\mathbf{b}^{(n)} \in \mathbb{R}^{n_{emb} + C}$ are the trainable parameters of the module.

The output $\mathbf{O} \in \mathbb{R}^{(T+2) \times C}$ is then a matrix of probabilities of each code for each visit after the start visit built from the visit histories and each previous code in the same visit. Each code corresponds to a conditional probability in the product from Eq. (6). We train our model using the binary cross-entropy loss function over each medical code

(treating the problem as a multi-label classification problem) with masking applied such that the start visit as well as any padded visits (of all zeros) do not contribute to the loss. The architecture of our model is shown in Fig. 1.

## Additional features and considerations
Finally, We discuss different variants and add-on features of HALO.

**Conditional generation.** Our method generates electronic health record (EHR) data by using demographics $\mathcal{S}$ and chronic disease phenotypes $\mathcal{D}$ as labels, which are represented in our label vocabulary and applied to individual visits, as shown in Fig. 2. We selected these labels based on their relevance to downstream use cases. Each label is represented as a binary variable in $\mathbf{v}_l$, indicating the presence of the corresponding disease or demographics group indicator. These indicators are defined by concepts such as specific categories of genders, races, ethnicity, age groups, and more. We can easily extend this strategy to include other labels of interest, such as various biomarkers, patient outcomes, or even abstract patient embeddings.

**Unconditional generation.** Our setup generates electronic health record (EHR) data with conditional labels by incorporating a "label visit" in the data format, as illustrated in Fig. 2. This format enables easy generation of labeled and conditional data, which are highly valuable for using synthetic data in machine learning tasks and as an augmentation tool, particularly for rare cohorts. However, it's important to note that this formatting is optional. If desired, the "label visit" component can be removed from the EHR representation, and the architecture can be trained to generate unconditioned EHRs without any modification.

**Generation of continuous variables.** Our model can generate not only medical codes but also continuous variables, such as lab values and temporal gaps between visits. However, the availability of these additional variables in the generated data depends on their presence in the original dataset used for training. For example, the outpatient EHR dataset used in our study includes the time between visits, while the inpatient EHR dataset includes lab values.

In previous models, continuous values were typically generated using either GANs, which lack the autoregressive probabilistic modeling that we employ, or value predictors (such as time series analysis models), which we often found to produce average values with insufficient variance. To overcome these limitations, we model continuous variables within the healthcare domain by discretizing lab values and temporal gaps into clinically equivalent buckets. The resulting binary variables are included in the model's context, denoted as $\mathcal{C}$, before being converted back to continuous values through random uniform sampling within the corresponding bucket range. By using this approach, our model generates more realistic and diverse continuous variables than previous methods.

More specifically, to generate discrete versions of continuous variables, such as lab values and temporal gaps, we divide the range of each variable into several "buckets", as represented by the values $b_1, b_2, \cdots, b_{|l_j^{(t)}|}$, where $|l_j^{(t)}|$ refers to the number of buckets required. We determine the bucket ranges by either seeking advice from clinicians on practical ranges, creating granular but equivalent groupings, or using a histogram construction algorithm[45]. The same approach is applied to temporal gaps as well.

For example, the heart rate lab test with possible values ranging from 0 to 400 beats per minute down could be broken down into twenty different buckets splitting the overall span into smaller ranges that offer the same medical meaning for all their contained values. This breakdown could have $b_1 = (0, 40)$ and $b_7 = (90, 100)$. These buckets then convert the single continuous variable into many binary variables. Whenever the continuous variable is present in the original EHR, a

single one of those variables representing the corresponding bucket is set to 1 with the rest remaining 0. For instance, if a patient has a heart rate lab measurement of 93 bpm on their seventh visit, the seventh of the new heart rate variables would be 1 and the rest would remain 0. If there was no such lab measurement in the visit, they would all be 0.

These new binary variables are added to the wider code vocabulary $\mathcal{C}$ and treated in the same way as all of the other medical codes in the vocabulary by our HALO model during learning and generation. After generation, the specific lab values and inter-visit gaps are converted back into a continuous value by uniformly sampling from the corresponding bucket range at the very end.

This discretization allows us to maintain the same powerful and probabilistic modeling process, matching the probabilistic variance of real continuous values in the same way we match the variance of medical code presences. However, by building appropriately granular buckets, we can avoid losing meaningful information and maintain a full representation of a patient. We explore the performance of this approach further in our experiments.

### Reporting summary

Further information on research design is available in the Nature Portfolio Reporting Summary linked to this article.

### Data availability

The MIMIC-III inpatient EHR dataset[25] that we use is publicly available and may be downloaded and used freely after performing training and applying on physionet.org. Furthermore, we also released the synthetic data for each of our compared methods for both the inpatient and outpatient datasets at https://figshare.com/articles/dataset/HALO_Synthetic_Data/23811162. These datasets can then be used to reproduce the results and data statistics.

### Code availability

We make our code for the inpatient dataset experiments, including dataset construction, modeling building, training, and evaluation, available at https://github.com/btheodorou99/HALO_Inpatient[46]. Between this and the public availability of that dataset, all inpatient results can be fully reproduced. Furthermore, HALO is also included in the open-source machine learning package for healthcare PyHealth[47], where it is available for easy use in concert with various machine learning tasks.

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

## Author contributions

B.T. and J.S. proposed the method, B.T. and conducted all the experiments, B.T., C.X., and J.S. wrote the manuscript.

## Competing interests

The authors declare no competing interests.
