## [Peer Review File · Nature Communications]

Synthesize High-dimensional Longitudinal Electronic Health Records via Hierarchical Autoregressive Language ModelREVIEWER COMMENTS

Reviewer #1 (Remarks to the Author):

- The paper presents a new method called HALO that aims to produce high-dimensional synthetic longitudinal electronic health record (EHR) data. The approach involves using a multigranularity autoregressive approach with transformer decoder blocks to learn patient history embeddings, which are then utilized to generate synthetic data. The proposed method is evaluated on two real-world datasets, US claims data and MIMIC-III ICU stay dataset. Through a series of comprehensive experiments that compare it with two baselines and three state-of-the-art methods, the results demonstrate promising performance for synthetic data generation and downstream prediction tasks. Overall, the paper is well-written and well-organized. However, there are some issues that need to be addressed further, as outlined below:

- 1. As this paper highlighted that the proposed framework is designed for high-dimensional synthetic longitudinal EHR data generation, the authors should focus on discussing which modules or components are devised for handling the high-dimension issue.
- 2. The reviewer is curious about whether the final generation results are influenced by the position of the codes (as described in Equation 3) within the visit.
- 3. Lack of reference for the LSTM EHR Model. Is it proposed by the other researcher, or is a baseline designed in this study?
- 4. Comparing the results from HALO-Coarse, HALO, and LSTM in Tables 8 and 9, it seems that the performance improvement is mainly contributed by the transformer. Hence, it limited the contributions of the proposed method.
- 5. Lack of reference for the perplexity.
- 6. Show std (confidence intervals) in Tables 7 and 8.

Reviewer #2 (Remarks to the Author):

Yes, the work is noteworthy and significant in that it simultaneously addresses (1) high-dimensionality of feature relationships, (2) longitudinal coherency of the synthetic data (i.e., patient visits), and (3) continuous variables (e.g., vital signs and lab values) using a novel combination of techniques. The evaluation of the data, quality checking, and privacy analysis is also thorough.

The methodology is well described with formal notations and accessible explanatory text.

Regarding reproducibility... while the methodology and steps are very well-described...

Is the real input data available? Yes, but apparently only the MIMIC portion, which is addressed in the “Data Availability” section and the MIMIC data is easily obtained. The real-world outpatient data is not available, as expected, but this makes that portion of the world non-reproducible.

Is the synthetic data available? The “Data Availability” section only addresses the input data (i.e., MIMIC), but if a purported benefit of synthetic data is that it is a safe alternative that can be shared without privacy, security, and legal constraints – then where can the public access the synthetic data the generated by the authors?

Where is the public code repository that can be used to access HALO and the code used, in order to reproduce the authors experiments?

My fundamental objection to nearly all papers that make claims about synthetic data generation is that they cannot answer “yes” to those three questions.

The paper and results are an accomplishment, but how could someone reproduce it?

Minor issue: the interactive visualization “vega.github.io” link in Section 4.4 did not function with several different web-browsers, each reporting “Failed to decompress URL. Expected a specification, but received”.

Response to Reviewers:

We have updated our manuscript based on your comments. Below we provide point-to-point responses to your questions.

Reviewer #1

Q1. As this paper highlighted that the proposed framework is designed for high-dimensional synthetic longitudinal EHR data generation, the authors should focus on discussing which modules or components are devised for handling the high-dimension issue.

A1. The proposed framework includes two modules that address the high-dimensionality of synthetic longitudinal EHR data in different ways. Our visit-level modeling module utilizes transformers as the backbone of our model, which have demonstrated great effectiveness in handling high-dimensional and sequential data. For intra-visit, code-level modeling, the high-dimensionality poses a huge challenge to performing typical sequential modeling within memory. And we employ a masked linear layer to achieve efficient modeling even in challenging high-dimensional settings. We added a more focused discussion of these components to our paper to clarify their usage and benefits.

Q2. The reviewer is curious about whether the final generation results are influenced by the position of the codes (as described in Equation 3) within the visit.

A2. In this manuscript, we conduct our experiments with a random ordering of codes (i.e., we randomly shuffled the set of all codes before assigning code indices). To investigate the influence of code position on the final generation results, we also explored using different code ordering, including (1) prevalence-based ordering that we sorted the codes in descending order of their prevalence across all visits and (2) category-based ordering that we grouped codes by their respective categories and sorted alphabetically within each category. The experiments found no significant difference in results and demonstrated that the final generation results are not influenced by the ordering of the codes within the model. We have included an exploration of the impact on EHR modeling results on the inpatient dataset from 4 different orderings in our supplementary material.

Q3. Lack of reference for the LSTM EHR Model. Is it proposed by the other researcher, or is a baseline designed in this study?

A3. There are many LSTM-based predictive models for EHR based disease detection and progression modeling. While some previous works have used generative LSTM models to create unstructured medical text data, to the best of our knowledge, no approach has yet been proposed to use LSTM-based models for synthetic EHR generation. In this manuscript, we adapted “Natural language generation for electronic health records” (Lee 2016) to generate structured multi-hot visits rather than unstructured clinical notes, and we have added a citation to the paper.

Q4. Comparing the results from HALO-Coarse, HALO, and LSTM in Tables 8 and 9, it seems that the performance improvement is mainly contributed by the transformer. Hence, it limited the contributions of the proposed method.

A4. From the model design perspective, the HALO first introduces the transformer as one building block of the EHR generation task. From the performance perspective, while it is true that the transformer plays a significant role in improving performance, it is important to note that HALO-Coarse, which only uses the transformer, is not able to model intra-visit patterns such as medical code co-occurrences. The tasks in Tables 8 and 9 do not rely on this, but the inpatient results in Table 3 and code plots in Figure 3 demonstrate the limitations of HALO-Coarse (that only uses the transformer) and the contributions of the full proposed method. In these evaluations, HALO outperforms both HALO-Coarse and LSTM, demonstrating the effectiveness of the proposed model design. Specifically, HALO improves HALO-Coarse's 0.343 predictive F1 Score on the inpatient dataset by 21% to 0.414 in Table 3 and then HALO-Coarse's outpatient same visit bigram probability R^2 Score by 37% from 0.661 To 0.908.

Q5. Lack of reference for the perplexity.

A5. Perplexity is a common evaluation metric for generative modeling and language modeling in particular. We adopted it for synthetic EHR generation from there, and we have added a citation of a popular paper using perplexity and a discussion of its introduction to our task.

Q6. Show std (confidence intervals) in Tables 7 and 8.

A6. Thank you for pointing it out. We have added the standard deviation as confidence intervals in both tables and a comment in the manuscript.

Reviewer #2

Q1. Regarding reproducibility... while the methodology and steps are very well-described...Is the real input data available? Yes, but apparently only the MIMIC portion, which is addressed in the "Data Availability" section and the MIMIC data is easily obtained. The real-world outpatient data is not available, as expected, but this makes that portion of the world non-reproducible.

A1. Yes, the MIMIC inpatient dataset is easily obtained and allows full reproducibility of all corresponding results (we have added code and synthetic data to aid in that reproducibility). The outpatient data is not accessible, and we are aware of the limitation that it creates. However, the different, more longitudinal perspective it provides is valuable, and its lack of availability is why we also made sure that we had one publicly available dataset.

Q2. Is the synthetic data available? The “Data Availability” section only addresses the inpatient data (i.e., MIMIC), but if a purported benefit of synthetic data is that it is a safe alternative that can be shared without privacy, security, and legal constraints – then where can the public access the synthetic data generated by the authors?

A2. Thank you for raising this point. We have made each of the synthetic datasets (both HALO’s and each of the baseline datasets as they all passed each privacy evaluation) for both of the inpatient and outpatient dataset publicly available at https://drive.google.com/file/d/107ZzEa_meJEfS07ktuy8c4wj0YsMpCkV/view?usp=share_link and added the same link to our data availability section. This data can be used freely and reproduce the statistics and metrics that do not rely on the real data.

Q3. Where is the public code repository that can be used to access HALO and the code used, in order to reproduce the authors experiments?

A3. Thank you again for bringing this to our attention. We have now released all the code used to perform the MIMIC experiments at https://github.com/btheodorou99/HALO_Inpatient and added the link to the code availability section to allow for full reproducibility. Furthermore, we are in the process of integrating HALO into the popular open-source machine learning in healthcare package PyHealth where it will be made available for easy usage. We hope to have that completed by the end of the month.

Q4. My fundamental objection to nearly all papers that make claims about synthetic data generation is that they cannot answer “yes” to those three questions. The paper and results are an accomplishment, but how could someone reproduce it?

A4. Thank you for your thoughtful question about reproducibility. We agree that reproducibility is critical for scientific research, and we recognize that it can be especially challenging in the medical domain, where privacy concerns often limit data availability. In our paper, we have taken steps to improve reproducibility by making both the code and synthetic data available to the public. Additionally, we utilized at least one publicly available dataset to help ensure that our results can be reproduced. We hope that our efforts to make the code and synthetic data available will enable others to build upon our research and further advance the field.

Q5. Minor issue: the interactive visualization “vega.github.io” link in Section 4.4 did not function with several different web-browsers, each reporting “Failed to decompress URL. Expected a specification, but received”.

A5. Thanks for pointing it out. We discovered that the URL was too long (over 300,000 characters) and this caused it to break. To address this issue, we’ve shortened the URL by randomly selecting 1,000 points from the original plot, which had approximately 10,000 points. We believe this solution has fixed the issue, but if the URL breaks again, we have a backup plan: we can either provide a text file containing the full plot configuration or replace the plot with an even smaller subset of points. The current URL is still around 50,000 characters long.

REVIEWERS' COMMENTS

Reviewer #1 (Remarks to the Author):

I don't have any new comments. The authors have addressed my concerns.

Reviewer #2 (Remarks to the Author):

Thank you for addressing my comments and providing source code and synthetic data output. Transparency and reproducibility in science is paramount.

- The synthetic data does not include, and the source code does not cover, replicating continuous variables (section 4.6): gaps in days between visits, and lab values.
- The synthetic data is available in the python "pickle" format. That is the authors prerogative. For what it is worth, I converted this data into JSON and it was significantly smaller (in bytes) and readable by a variety of programming languages.
- The README with the source code contains no useful instructions.
- The synthetic data archive had no README whatsoever.

Presumably the authors want these resources to be reused. You should take the time to put them into shape where they can be reused.

Response to Reviewers:

The synthetic data does not include, and the source code does not cover, replicating continuous variables (section 4.6): gaps in days between visits, and lab values.

- Thank you for bringing this to our attention. We have added the code for handling both gaps and lab values in accordance with the contents of the inpatient dataset

The synthetic data is available in the python "pickle" format. That is the authors prerogative. For what it is worth, I converted this data into JSON and it was significantly smaller (in bytes) and readable by a variety of programming languages.

- Thank you for taking the effort to convert the data. We have converted the data to JSON format as well and uploaded a new version of the .zip file at https://drive.google.com/file/d/107ZzEa_meJEfS07ktuy8c4wj0YsMpCkV/view?usp=share_link.

The README with the source code contains no useful instructions.

- We have added to the README with step-by-step information about how to run the experiments in an end-to-end manner.

The synthetic data archive had no README whatsoever.

- We have included a README file in the top level of the .zip file with information about the structure of the folder, the format of the data, and how to use the data.